# The Unique Identification of an Unknown Soldier from the Estonian War of Independence

**DOI:** 10.3390/genes12111722

**Published:** 2021-10-28

**Authors:** Anu Aaspõllu, Raili Allmäe, Fred Puss, Walther Parson, Küllike Pihkva, Kairi Kriiska-Maiväli, Arnold Unt

**Affiliations:** 1Department of Nutrition Research, National Institute for Health Development, 11619 Tallinn, Estonia; 2Archaeological Research Collection, Tallinn University, 10130 Tallinn, Estonia; rallmae@gmail.com; 3Faculty of Arts and Humanities, University of Tartu, 50090 Tartu, Estonia; fred@isik.ee; 4Department on Language History, Dialects, and Finno-Ugric Languages, Institute of the Estonian Language, 10119 Tallinn, Estonia; 5Institute of Legal Medicine, Medical University of Innsbruck, 6020 Innsbruck, Austria; walther.parson@imed.ac.at; 6Forensic Science Program, The Pennsylvania State University, University Park, PA 16802, USA; 7Preservation Department in Tartu, National Archives of Estonia, 50411 Tartu, Estonia; Kyllike.Pihkva@ra.ee; 8Document Department, Estonian Forensic Science Institute, 13419 Tallinn, Estonia; kairi.kriiska-maivali@ekei.ee; 9Estonian War Museum-General Laidoner Museum, 74001 Tallinn, Estonia; arnold.unt@esm.ee

**Keywords:** human remains identification, anthropological assessment, genealogy/kinship analysis, DNA typing, Y-chromosomal STR haplotypes

## Abstract

The identification of human remains is challenging mostly due to the bad condition of the remains and the available background information that is sometimes limited. The current case report is related to the identification of an unknown soldier from the Estonian War of Independence (1918–1920). The case includes an anthropological study of the remains, examinations of documents found with the exhumed remains, and kinship estimations based on archival documents, and DNA analyses. As the preliminary data pointed to remains of male origin, Y-chromosomal STR (short tandem repeat) analyses of 22 Y-STR loci were used to analyze the exhumed teeth. Reference samples from individuals from two paternal lineages were collected based on archival documents. Y-chromosomal STR results for the tooth samples were consistent with a patrilineal relationship to only one reference sample out of two proposed paternal lineages. Based on the provided pedigrees in the consistent case, the Y-STR results are approximately four million times more likely if the tooth sample originated from an individual related along the paternal line to the matching reference sample, than if the tooth sample originated from another person in the general population. Special considerations have to be met when limited evidence is available.

## 1. Introduction

The Republic of Estonia became independent through the Estonian War of Independence (28 November 1918–2 February 1920). Around 100,000 people out of one million Estonians participated in the war, of those, 5889 were lost through death, and 433 were captured or went missing in action [1]. The identified dead bodies were buried in homeland cemeteries.

The number of unidentified victims forced the Estonian Police Headquarters to start the examination of field graves in 1922. The preserved documents reveal that the graves were in poor condition with many unidentified persons. On 19 June 1925, the Riigikogu (Estonian Parliament) approved the Law on the Maintenance of the Remains of Estonian Soldiers who died during the Estonian War of Independence. However, the implementation of the corresponding law did not improve the situation with the war graves or the identification [2,3,4]. After the year 1925 exhumations not performed. 

The identification of the victims began again over half a century later, which nowadays needs contributions from many fields of science. Here, we present one of the cases of the identification of an Estonian unknown soldier. The remains of the soldier were exhumed from Võru County on 16 September 2017. 

## 2. Materials and Methods

### 2.1. Exhumation of Remains

The exhumation was carried out by the Estonian War Museum pursuant to § 8 of the War Graves Protection Act passed in 2007 (reburial from an inappropriate place). The origin of the grave was known from locally inherited information and a 1922 police report [3,4].

The unknown soldier was exhumed from a cross-marked grave located in the hilly-forested landscape of southeast Estonia. The location of the grave was one meter above the swampy area surrounding it. Typical for the soil type (Podzol), the environment at the grave was acidic. On opening the grave, a skeleton was identified lying on his supine, slightly gravitated to his left, with his arms folded over his head. The depth of the grave was 65 cm; the highest point of the skeleton was 40 cm below ground level. A remnant of the wallet found with the human remains contained a couple of coins and folded paper money in addition to fragments of paper documents. Besides a dozen Mosin-Nagant M1891 rifle cartridges were found with the remains.

### 2.2. Anthropological Analysis of Skeletal Remains

The putative age at death and biological sex of the exhumed remains were assessed based on common standards and methods [5,6,7,8]. The biological sex was assessed according to morphological features on skull and innominate bones [5]. Additionally, osteometric measurements were performed [9,10] in order to assess the biological sex. The epiphyseal fusion of long bones, innominate and clavicle, the closure of cranial sutures and the morphology of auricular surface of innominate bones were evaluated to estimate the age at the death [6,7,8]. The symphyseal surface of pubic bone was not available. The calculated body weight and the body height were performed according to formulas Ruff et al. [11].

### 2.3. Conservation and Identification of Two Paper Documents 

The paper content of the wallet was separated into two parts. The conservation was carried out based on the principle of reversibility and conservation ethics [12,13]. Ferrous corrosion was removed and the fragments of the two documents were washed, sized and the fragments reassembled and supported (oxalic acid, methyl cellulose gel, Gore-Tex^®^ sheets (W. L. Gore & Associates Inc., Newark, DE, USA), ethanol, water, blotting paper, Hollytex^®^ (Kavon Filter Products Co, Farmingdale, NJ, USA); lined using light tissue paper, missing fragments were replaced using toned Japanese conservation paper Mulberry Thin; adhesives: diluted methyl cellulose and wheat starch paste). The documents were buffered (Bookkeeper^®^ Deacidification Spray (Preservation Technologies L.P., Cranberry Township, PA, USA), magnesium oxide) using common methods in the practice of paper conservation. As a result of the conservation, the physical integrity of the documents was restored, as well as the determination of the script on the two conserved documents. Ensuring the best storage life of extremely fragile documents, both were enclosed by a specially designed protective enclosure made of archival quality cellulose materials. For text detection, a VSC 6000 video spectral comparator (Foster + Freeman Ltd., Evesham, Worcestershire, UK) was used.

### 2.4. Identification of Putative Genealogy of Remains

The church and vital statistics documents from the National Archives of Estonia were used [14,15,16].

### 2.5. DNA Analyses 

Two teeth (1 molar, 1 canine) were analyzed, as well as two buccal swabs as reference samples (two male samples—Testee 1 and Testee 2). The tooth samples were cleaned and purified prior to drilling. The tooth powder was subjected to lysis and DNA was extracted according to the protocol in Bauer et al. [17]. To identify the Y-chromosomal haplotype, the following Y-chromosomal STR-systems were amplified: DYS576, DYS389 I, DYS448, DYS389 II, DYS19, DYS391, DYS481, DYS549, DYS533, DYS438, DYS437, DYS570, DYS635, DYS390, DYS439, DYS392, DYS643, DYS393, DYS458, DYS385, DYS456, and YGATAH4 using PowerPlex^®^ Y23 System (Promega Corporation, Madison, WI, USA). The negative and positive control samples were included. The amplified STR products were separated using capillary electrophoresis (AB 3500 XL; Thermo Fisher Scientific Inc., Waltham, MA, USA). Kinship analysis was based on the assumed family pedigrees obtained from the archival records depicted in Figure 1 and performed using the Y-Chromosome STR Haplotype Reference Database (YHRD; www.yhrd.org; R58; accessed on 19 March 2020) with the following settings: Eastern European subpopulation (n:48,028 haplotypes); number of transmission events between Testee 1 and Tooth 1: 7; frequency estimation method: Discrete Laplace; calculation method: Consider only one-step mutations per transmission event. 

## 3. Results

### 3.1. Anthropological Analysis of the Skeletal Remains

The investigation of the skeleton connected remains firstly to a young adult male whose biological age at death was most likely under 30 years, but definitely above 20 years. The age range was concluded based on open cranial sutures, fused spheno-occipital synchondrosis, fused epiphyses of long bones, youthful appearance (25−29 years) of the auricular surface of innominates and complete fusion of epiphyses of clavicles.

The first assumption of the male skeleton was based on the lengths of long bones of upper and lower extremities and relatively large skull. However, more detailed examination and scoring of morphological traits of the skull and of the innominate bones revealed relatively high proportion of feminine traits (Table 1). The available eight measurements of length, width and circumference of the long bones of upper and lower extremities were also not very helpful in biological sex determination (Table 2). To estimate biological sex the sum of diagnosic coefficents (DK) of sex discriminating features should exceed certain limit. At probability *p* = 0.05 the limit is ±128; positive value refers to female sex and negative refers to male sex [10]. For the skeleton in the current study the DK value is −14. Thus, based on eight available osteometric features, the biological sex is inconclusive (Table 2). 

The investigated traits of the skeleton do not convincingly refer to male or female sex of a deceased individual. The skeleton may belong to a tall masculine woman, but at the same time, it may belong to slender built (gracile), and of a medium body height man. The calculated body height and body weight from femoral bone would be 173.41 cm/62.1 kg for male skeleton; and for female skeleton 172.68 cm/64.47 kg.

According to Juhan Aul [18] the average male body height was 172.03 cm (variation 150−195 cm in Estonia before World War II, and the average body weight was 69.67 kg (variation 48−100 kg). In the beginning of the 20th century, the average female body height in Estonia [19] was 161.83 cm (variation 145.3−181.0 cm), and the average body weight 61.6 kg (variation 43.0−91.0 kg).

### 3.2. Document Identification

The contents of the wallet were to some extent preserved probably due to the iron and copper details of the wallet and the silver content of the coins. The corrosion had turned the folded paper into a monolithic mass protecting it from decay. Unfolding the folded square rusty lump of approximately 4 × 5 cm during the conservation resulted in the estimation of two extremely fragile and friable paper documents in addition to paper money and two coins. These paper documents were severely corroded and showed significant losses and faded text. 

The conservation and restoration made it possible to determine the script on the two paper documents, both on the thin pulp machine-made paper: “Lubatäht” (Permission slip, 10.8 × 16.9 cm) and “Tunnistus” (Certificate, 10.7 × 18.3 cm). The media identified on the paper is as follows: relief print, iron gall ink, stamp ink (Figure 2 and Figure 3, respectively).

Even though the documents were highly damaged, the preserved area on the left side was less damaged, and the stamp impression was detectable using a VSC 6000 video spectral comparator in the infrared luminescence condition, which provided information about the putative issuer of the document (Figure 4). 

The pivotal stamp impression was compared to a document from the National Archives of Estonia that revealed the same stamp image as well as names pointing to Jaan Luts (male), N.N. Altin and Ida Peterson (female) (Figure 5).

Due to the stamp impression referring to Tartu County the document in Figure 5 was found at the National Archive of Estonia [20], which is the handwritten copy of the Document at Figure 3.

### 3.3. Molecular Genetic (Y-Chromosomal) Analyses

One tooth sample gave successful results in all 22 Y-chromosomal STR loci investigated. Another tooth resulted in 18 of the 22 Y-chromosomal STR loci investigated. The two haplotypes yielded identical results in the successfully analyzed overlapping Y-chromosomal STR loci. 

The haplotype of Testee 1 matched the Y-STR haplotype obtained for sample Tooth 1.

The likelihood ratio of a patrilineal relationship between Testee 1 and Tooth 1 versus a non-relationship was approximately 4,015,733; therefore, providing very strong evidence for relatedness between Testee 1 and Tooth 1 along the paternal lineage.

The sample from Testee 2 had discrepancies (one-step differences) in three Y-STR loci: DYS389II, DYS385, YGATAHA4.

The likelihood ratio of a patrilineal relationship between Testee 2 and Tooth 1 versus a non-relationship was approximately 7; therefore, providing very weak evidence for relatedness between Testee 2 and Tooth 1 along the paternal lineage.

## 4. Discussion

The excavated human remains were completely skeletal. The skeletal remains belong to the young adult whose biological sex cannot be determined by the anthropological methods. Besides the skeleton, the remnants of pocket areas of the soldier’s coat were found, which consisted of a dozen Mosin-Nagant M1891 rifle cartridges in the remains of his right pocket, and copper sleeves and metal parts of the leather wallet in his left pocket.

Based on the paper documents found in the wallet remnants, the archival document from the National Archives of Estonia [20] points to the same names, but the name of Karl Kenner (male) has been crossed out in the archival document, and those of Jaan Luts (male) and Altin have been written instead (Figure 5). The name Ida Peterson (female) has remained unaltered. Another preserved document in the remnants was issued to Elga Grünberg (female). Biographical analyses of the named persons (Elga Grünberg, Mr or Ms Altin and Jaan Luts) disclosed that there was one Jaan Luts, who could be the person buried in the grave in question [21].

Therefore, for the identification of the exhumed remains, the pedigree of Jaan Luts was considered. He had two brothers, one of them died in 1904 in the Russo-Japanese War, the other brother had two sons who have altogether three daughters (all alive) but had no sons. The two closest living male-line relatives, Testee 1 and Testee 2, were from a common ancestor Jüri Luts (1788–1879), whose documented relationship is shown in Figure 1 [National Archives of Estonia: Soul revision lists, church books, registers of commune members, vital statistics documents]. 

The Y-chromosome testing performed matched the sample of Testee 1 with that of the exhumed remains, thus making it possible to associate the latter one to Jaan Luts (1887–1919), (It is not without interest to note that in 1940 the body was identified as that of Johannes Karu (aka Karo), based on the testimonies “without any doubt” from the latter’s sister and her husband, who established matches in the descriptions about the deceased’s age, height, hair, teeth, clothing etc [National Archives of Estonia: ERA.2124.2.151, L 69-78]). Based on the provided pedigree, the Y-STR results are approximately 4 million times more likely if the tooth sample originated from an individual related along the paternal line to a particular reference sample, than if the tooth originated from another individual in the general population. The sample from Testee 2 showed only a very weak indication that he could be related to the soldier (likelihood ratio in the same conditions approximately 7). The genetic pedigree discrepancies (Figure 1) using actual DNA analysis results could, on the one hand, be explained by a cuckoldry event. There is no research yet performed (or at least published) about cuckoldry (also referred to as non-paternity, extra-pair paternity, misattributed paternity) in Estonia; that is, the correspondence of official vital statistics records as compared to biological (DNA-based) relationships. Corresponding data is available from many other parts of the world. One of the first studies in this field was published in 2001 about an Argentinian village Aicuña [22]. The results showed an 85% match between conventional and molecular genealogies over the documented 12 generations. In 2009, King and Jobling [23] researched 40 British surnames and among five where there was most likely one common ancestor, the study suggested a non-paternity rate of 1–4.5%. According to the authors, the real non-paternity rate would be even lower because of the possibility of multiple founders of the surname, but only one survived lineage.

For surnames with unprovable documented genealogy, not only the non-paternity has to be considered, but also other events like adoption or fostering and matrilinear transmission. When researching Catalan surnames, Solé-Morata et al. [24] found the introgression rate of Y-chromosomes into a surname through those events to be 1.5–2.6%.

Based on such single studies in several parts of the world, Larmuseau [25,26] has contradicted the cuckoldry rates presented in the scientific literature and among behavioral scientists as being 10–30% and ascertains that the rate is not higher than 1–2% per generation either historically or among the present-day population. Only in some human populations with very specific characteristics has the rate been much higher; for instance, among South American Yanomamis with 10%. Larmuseau also points out that in the Mexican population, the non-paternity rate is 20% higher among families of low socioeconomic status and that additional studies are necessary to create a wider picture of human cuckoldry behavior and the factors that influence it.

There are six generations between Jaan Luts and Testee 2. In this case, with the mean cuckoldry rate of 6–12%, the probability of finding a cuckoldry case (as there is no adoption or matrilinear transmission of the surname in the family) is 1:8–1:16. Although the present case is statistically not representative, with the result of 1:2, it provides an interesting clue for further studies of the matter in Estonia. 

As the Y-haplotype analyses excluded one pedigree, the attention has to be paid to mutation rates, which are reportedly 65 per 15,021 individuals for the DYS389II locus, 70 per 27,911 individuals for the DYS385 locus, and 27 per 8971 individuals for the YGATAHA4 locus (www.yhrd.org; Accessed on 19 March 2020) [27].

As this case illustrates, ethical matters have to be considered very carefully in the present field. On the one hand, there is no way of knowing which generation DNA mutation rates and cuckoldry events could be involved, and on the other hand, regardless of the possible distance in generations the sensitivity of people towards the liberality and morality issues of their ancestors varies greatly.

## 5. Conclusions

DNA-based methods for sex determination can confirm biological sex if results of anthropological analysis do not convincingly refer to male or female origin of exhumed skeleton.

Y-chromosomal STR analyses yielded successful and identical results in the overlapping typed Y-STR loci for two investigated tooth samples from the exhumed remains from the Estonian War of Independence, which enabled identification the unknown soldier and somehow to resolve preliminary controversial evidence.

The Y-chromosomal STR results of tooth samples are consistent with a patrilineal relationship to only one reference sample out of two proposed paternal lineages. Based on the provided pedigrees in one of the cases, the Y-STR results are approximately 4 million times more likely if the sample of the tooth originated from an individual related along the paternal line to the particular reference sample, than if the tooth originated from another, paternally unrelated individual in the general population. Another pedigree was questionable.

Using published family trees for human identification purposes the careful control must be implemented to rule out possible mistakes due to the biological non-paternity.

Special considerations have to be met when only limited evidence is available. However, importance of integrated scientific approaches is utmost relevant.

## Figures and Tables

**Figure 1 genes-12-01722-f001:**
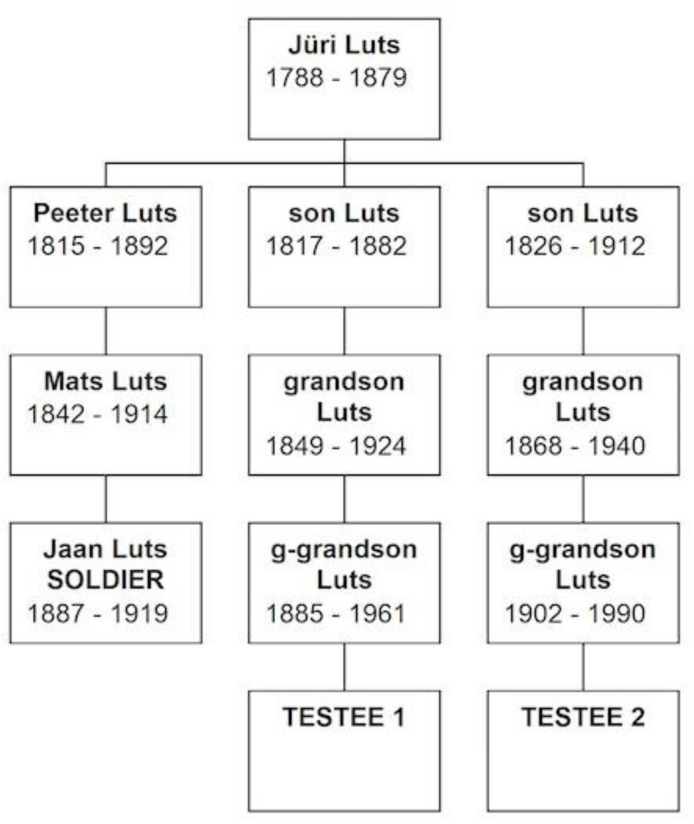
The descendant chart for the identification of an unknown soldier.

**Figure 2 genes-12-01722-f002:**
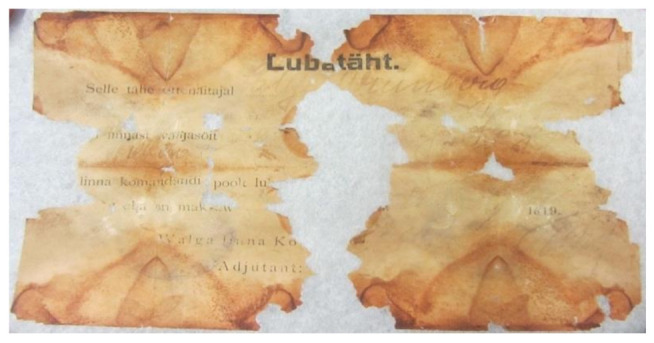
“Lubatäht” (Permission slip).

**Figure 3 genes-12-01722-f003:**
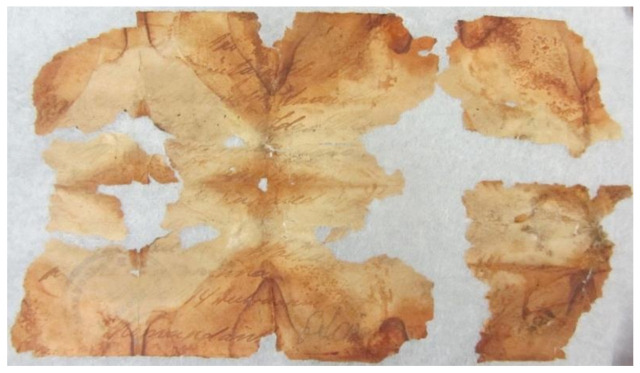
“Tunnistus” (Certificate).

**Figure 4 genes-12-01722-f004:**
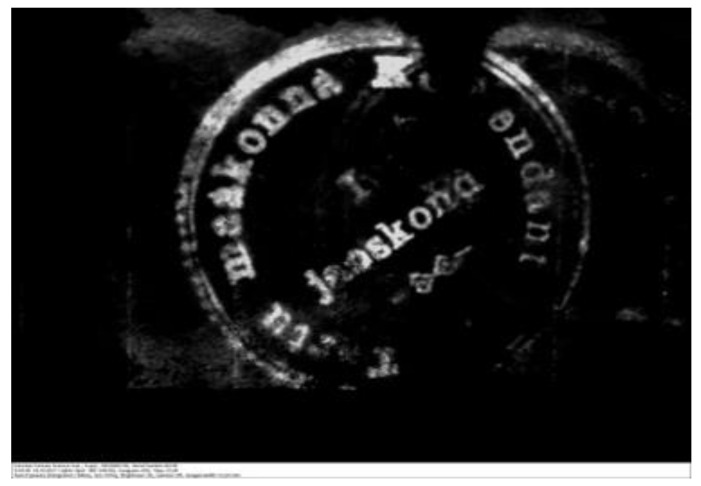
The Stamp impression with the text “Tartu maakonna Komandant” (Commandant of Tartu County).

**Figure 5 genes-12-01722-f005:**
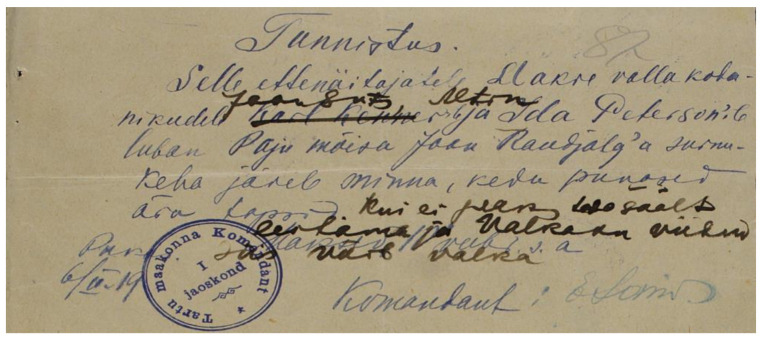
The document from the National Archives of Estonia pointing to the names Jaan Luts, N.N. Altin and Ida Peterson [20].

**Table 1 genes-12-01722-t001:** Assessment of biological sex based on morphological traits [5].

The Site	Period			Skeleton No	
Võru County, Estonia	Estonian War of Independence			1	
Diagnostic traits (WEA 1980)	♂				♀
**Cranium**					
Glabella	+2	+1	0	−1	−2
Superciliary arch	+2	+1	0	−1	−2
Frontal tubera	+2	+1	0	−1	−2
Frontal inclination	+2	+1	0	−1	−2
Mastoid process	+2	+1	0	−1	−2
Nuchal plane	+2	+1	0	−1	−2
External occipital protuberance	+2	+1	0	−1	−2
Supraorbital margin of frontal bone	+2	+1	0	−1	−2
Shape of the orbit	+2	+1	0	−1	−2
Supramastoid crest	+2	+1	0	−1	−2
**Mandible**					
General view	+2	+1	0	−1	−2
Mentum	+2	+1	0	−1	−2
Angle	+2	+1	0	−1	−2
Inferior margin	+2	+1	0	−1	−2
**Innominate**					
Preauricular sulcus	+2	+1	0	−1	−2
Greater sciatic notch	+2	+1	0	-1	−2
Arc compose	+2	+1	0	−1	−2
Ischial body	+2	+1	0	−1	−2
General traits and shape on innominate	+2	+1	0	−1	−2

**Table 2 genes-12-01722-t002:** Biological sex assessment from osteometric traits.

Measurement (mm)	Female—Equal or Below (mm)	Male—Equal or Above (mm)	Measured Value (mm)	Diagnostic Coefficient (DK)	Estimated Sex
Martin & Saller [9]	Garmus & Jankauskas [10]				
FEM 2 physiological lenght	442	443	475	−111	male
FEM 20 circumference head	145	146	146	−40	male
FEM 21 distal epiphysis width	78	79	75	+128	female
FEM 8 circumference middle	85	86	92	−159	male
HUM 4 distal epiphysis width	61	62	59	+7	female
HUM 7a circumference middle	66	67	63	+86	female
TIB 3 proximal epiphysis width	76	76.1	74	+93	female
TIB 10a circumference *foramen nutricium*	89	89.1	93	−18	male

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
