# Peer review of "The Unique Identification of an Unknown Soldier from the Estonian War of Independence"

_genes, 2021, doi:10.3390/genes12111722_

Round 1

Reviewer 1 Report

The authors responded to my suggestions appropriately.

Author Response

Point 1: Are the results clearly presented?  (Can be improved) â€¨

Response 1: We do agree with the reviewer as every revision helps to improve the manuscript.   

Reviewer 2 Report

In this case report, authors have used different techniques to identify remains and checked with Testee 1 & 2. It’s a well written case report …

The authors have a typo error in line #133. Kindly resolve / reword it

Author Response

Point 1: The authors have a typo error in line # 133. Kindly resolve/revord it.ʉ۬

Response 1: The double phrase "would be" corrected (one "would be" deleted).   

This manuscript is a resubmission of an earlier submission. The following is a list of the peer review reports and author responses from that submission.

Round 1

Reviewer 1 Report

This case report covers several kinds of approaches for identifying skeletal remains. The techniques of the physical anthropology and the forensic document analysis are introduced here as well as of the forensic DNA analyses so that this report is valuable as forensic case report.

However, this report needs several revisions. Followings are the points to be considered.

P2, L72-:

Two much sentences were described in a round bracket. It is not clear to know the process for reassemble and support the paper sheet. Same problem is found in the next round bracket.

The authors should describe the methods carefully.

P2, L84:

It is not clear the origin of the female buccal swab even after reading the descriptions on Fig. 5. The authors should mention the origin of this female sample even though it was not appropriate for DNA tests in this case.

P3, L120-:

I am wondering why the stamp image could link the document in a wallet to the archived document (Fig. 4). I don’t think the stamp is not so very unique to find one documents in archives. The archived document is a duplicate of the one in a wallet?

Did the infrared observation reveal the faded text clearly and found some written name in the paper? If so, the image under infrared with written names should be shown.  

The authors should mention the relationship between folded corroded paper and Fig.4 clearly.  

Title:

Overall, I couldn’t find remarkable mysterious contents in the report. The author had better delete “one mystery resolved” from the title.       

Author Response

This case report covers several kinds of approaches for identifying skeletal remains. The techniques of the physical anthropology and the forensic document analysis are introduced here as well as of the forensic DNA analyses so that this report is valuable as forensic case report.

 However, this report needs several revisions. Followings are the points to be considered.

 P2, L72-:

Two much sentences were described in a round bracket. It is not clear to know the process for reassemble and support the paper sheet. Same problem is found in the next round bracket.

The authors should describe the methods carefully.

Answer: Agree and text corrected.

 P2, L84:

It is not clear the origin of the female buccal swab even after reading the descriptions on Fig. 5. The authors should mention the origin of this female sample even though it was not appropriate for DNA tests in this case.

Answer: Female sample was collected based on genealogical data for sake of analysis of autosomal markers, however in the current context it is not relevant, so the text is corrected.  

P3, L120-:

I am wondering why the stamp image could link the document in a wallet to the archived document (Fig. 4). I don’t think the stamp is not so very unique to find one documents in archives. The archived document is a duplicate of the one in a wallet?

Did the infrared observation reveal the faded text clearly and found some written name in the paper? If so, the image under infrared with written names should be shown.  

The authors should mention the relationship between folded corroded paper and Fig.4 clearly.  

Answer: The stamp image is a pivotal evidence in linking Estonian region for searches in archival information about the putative issuer of the document and linking the document with putative individuals as mentioned in the text. Text modified.

Title:

Overall, I couldn’t find remarkable mysterious contents in the report. The author had better delete “one mystery resolved” from the title.       

Answer: Agree, the title modified.

Reviewer 2 Report

Manuscript Review

Genes - February 2021

Identification of an unknown soldier from the Estonian War of Independence – one mystery resolved

Overview:

This paper describes the identification of skeletal remains from the Estonian War using a combination of DNA-based (Y-STRS), anthropological, and document examination methods.

General Comments:

* Overall improvement of English in some parts are required to make it easier to read and to better understand the links between various statements and concepts.

* Some data and information in the Discussion should be moved into the results discussion.

* Anthropological methods and results were not clearly described.

* No new, novel, or more definitive DNA-based methods have been used in the case study to demonstrate the strength or power of DNA typing.   But rather, the Y-STR typing highlighted two things:

               1)  DNA-based sex determination is more powerful than anthropological methods for sex determination (particularly when the results for the assignment of sex using skeletal methods are not strongly definitive, such as in this case).

               2)  Y-STR typing can detect non-paternity (compared to published family trees), highlighting the concerns, potential problems and limitations when using archival records for HID purposes.

Specific Comments:

Abstract

*Lines 25-26 refer first to using Y-chromosomal markers and then say 22 STR loci were used in the same sentence.  This could initially be confused to think that both Y-STRs and autosomal STRs were used in this case.   Consider changing to simply “22 Y-chromosomal STR loci…….”

Introduction

* Good background information about the Estonian conflict and why this skeleton was exhumed for identification.

* No mention at all about how an identification from skeletal remains would traditionally be made, and using the various methods employed in this case study.   A brief background and summary of the relevant DNA-based methods used for the ID of historical remains, anthropological approaches for age, sex, height etc (creating a biological profile), and document examination is warranted (including key references for each).

Materials & Methods

* Line 59:   Use correct anatomical term:  “lying supine” rather than “ lying on his back”

* There is no mention of the skeletal inventory (which bones in the skeleton were recovered or not)

* Line 65-67.   These references to the methods used to age and sex the skeletal remains are not the original publications/methods/formulae/casts that the methods were based on, but rather a mixture of recommendations, textbooks, seminar proceedings and field manuals.   I would ask the authors to consider briefly describing the methods (and bones used for each test) and also including the original publication/description of the methods that were actually used in this study as there are several different methods that can be used to determine age and sex from a skeleton.

* Line 87 mentions DNA extraction was performed according to Ref 11 – however this paper is comparing several variations and extraction methods.  Which protocol did the authors use?  Where possible, always cite the original description/publication of a method used.

* No method for DNA quantification is described.

* Lines 88-92 list the Y-chromosomal STR markers or loci (not systems); however collectively together they form a Y-STR system/kit.   Was a commercial Y-STR kit (such as the Y-Filer kit) used here?  If so, then this needs to be described here.

* Line 93 refers to Figure 5.  This is the first figure cited in the paper, and therefore it should be listed as Figure 1.   However, if the authors prefer to leave Fig 5 in the same order (as the fifth figure in the body of the text), then consider revising the wording in the M&M section to be more general such as “Kinship analysis was based on assumed family pedigrees obtained from the archival records…..”.

Results

* Sex was initially based on long bones and a large skull; however the pelvis is a much more reliable indicator of sex.  Why was this bone not examined?  Were the Os coxa not recovered?

* It is not clear to me the connection between the a document in the archives with the same stamp as seen on one of the documents found with the skeleton, and the potential ID of the skeleton as Luts.   Just because the document in the archives with the same stamp has a name pointing to Jaan Lutz doesn’t mean the document with the skeleton also points to being Lutz just because the stamp is the same.     Please clarify this assumption to make it clear to the reader.

* Why even bother testing (and reporting) the negative result for the female reference sample when only using Y-markers? 

Discussion

* Lines 157-161.  Now the authors talk about more tests being performed not described in the Results section (innominate bones etc.) – describe all tests performed in the Results section, and then describe what they mean here in the discussion section.

* Lines 166-7.    Authors report the stature and weight estimates of the skeleton examined – these data should be in the Results section – and then how these estimates compare to historical records can then be discussed here in the Discussion.

* Lines 173-181 somewhat explain the connection between the papers and the skeleton, but because this is not clear in the results section it is confusing to the reader early on.    This connection needs to be made a lot clearer.

* Make a more succinct link between your discussion of surnames and non-paternity, and a clear line as to why it is important in this particular case.

* Lines 229-232. Make a clearer statement here about why Y-STR mutations rates do not explain the non-paternity result for Testee 2.

Conclusion

* Although the authors suggest that these remains have been positively identified – did they also explore the possibility of these bones belonging to any other missing soldiers from the same paternal lineage (with a concordant Y-haplotype)??

* Line 249.   What does “Another pedigree was questionable” mean?   Are the authors referring to the Testee 2 assumed pedigree?

* Rephrase the closing sentence – it does not make sense.

Tables

* Table 1 – what is the source of these metrics/criteria?  Cite them here.

* Table 2A – make sure the reference formatting is consistent for then all in the first column in the table.

Author Response

Identification of an unknown soldier from the Estonian War of Independence – one mystery resolved

Overview:

This paper describes the identification of skeletal remains from the Estonian War using a combination of DNA-based (Y-STRS), anthropological, and document examination methods.

General Comments:

 * Overall improvement of English in some parts are required to make it easier to read and to better understand the links between various statements and concepts.

* Some data and information in the Discussion should be moved into the results discussion.

* Anthropological methods and results were not clearly described.

* No new, novel, or more definitive DNA-based methods have been used in the case study to demonstrate the strength or power of DNA typing.   But rather, the Y-STR typing highlighted two things:

               1)  DNA-based sex determination is more powerful than anthropological methods for sex determination (particularly when the results for the assignment of sex using skeletal methods are not strongly definitive, such as in this case).

               2)  Y-STR typing can detect non-paternity (compared to published family trees), highlighting the concerns, potential problems and limitations when using archival records for HID purposes.

 Specific Comments:

 Abstract

 *Lines 25-26 refer first to using Y-chromosomal markers and then say 22 STR loci were used in the same sentence.  This could initially be confused to think that both Y-STRs and autosomal STRs were used in this case.   Consider changing to simply “22 Y-chromosomal STR loci…….”

Answer: Agree, corrected.

 Introduction

 * Good background information about the Estonian conflict and why this skeleton was exhumed for identification.

* No mention at all about how an identification from skeletal remains would traditionally be made, and using the various methods employed in this case study.   A brief background and summary of the relevant DNA-based methods used for the ID of historical remains, anthropological approaches for age, sex, height etc (creating a biological profile), and document examination is warranted (including key references for each).

Answer: By our idea the case study based on integrated data from different scientific fields for identification is provided in concise.

Materials & Methods

 * Line 59:   Use correct anatomical term:  “lying supine” rather than “ lying on his back”

Answer: Agree, corrected.

* There is no mention of the skeletal inventory (which bones in the skeleton were recovered or not).

Answer: The case study is based on the data of skeletal remains.

* Line 65-67.   These references to the methods used to age and sex the skeletal remains are not the original publications/methods/formulae/casts that the methods were based on, but rather a mixture of recommendations, textbooks, seminar proceedings and field manuals.   I would ask the authors to consider briefly describing the methods (and bones used for each test) and also including the original publication/description of the methods that were actually used in this study as there are several different methods that can be used to determine age and sex from a skeleton.

Answer: Agree, corrected.

* Line 87 mentions DNA extraction was performed according to Ref 11 – however this paper is comparing several variations and extraction methods.  Which protocol did the authors use?  Where possible, always cite the original description/publication of a method used.

Answer: Authors used a routine procedure for DNA extraction using Qiagen EZ1 Advanced instrument.

* No method for DNA quantification is described.

Answer:  Negative and positive control samples were included for elimination of false positive data.

* Lines 88-92 list the Y-chromosomal STR markers or loci (not systems); however collectively together they form a Y-STR system/kit.   Was a commercial Y-STR kit (such as the Y-Filer kit) used here?  If so, then this needs to be described here.

Answer: Yes, commercial kits was in use.

* Line 93 refers to Figure 5.  This is the first figure cited in the paper, and therefore it should be listed as Figure 1.   However, if the authors prefer to leave Fig 5 in the same order (as the fifth figure in the body of the text), then consider revising the wording in the M&M section to be more general such as “Kinship analysis was based on assumed family pedigrees obtained from the archival records…..”.

Answer: We would prefer to leave the Fig 5 in the same order.

Results

 * Sex was initially based on long bones and a large skull; however the pelvis is a much more reliable indicator of sex.  Why was this bone not examined?  Were the Os coxa not recovered?

Answer: Innominates were examined, the available pelvic traits are presented in Appendix  1.

* It is not clear to me the connection between the a document in the archives with the same stamp as seen on one of the documents found with the skeleton, and the potential ID of the skeleton as Luts.   Just because the document in the archives with the same stamp has a name pointing to Jaan Lutz doesn’t mean the document with the skeleton also points to being Lutz just because the stamp is the same.     Please clarify this assumption to make it clear to the reader.

Answer: The stamp image is a pivotal evidence in linking Estonian region for searches in archival information about the putative issuer of the document and linking the document with putative individuals as mentioned in the text. Text modified.

* Why even bother testing (and reporting) the negative result for the female reference sample when only using Y-markers? 

Answer: Female sample was collected based on genealogical data for sake of analysis of autosomal markers, however in the current context it is not relevant, so the text is corrected.  

  Discussion

 * Lines 157-161.  Now the authors talk about more tests being performed not described in the Results section (innominate bones etc.) – describe all tests performed in the Results section, and then describe what they mean here in the discussion section.

Answer: Tranferred to Results section

* Lines 166-7.    Authors report the stature and weight estimates of the skeleton examined – these data should be in the Results section – and then how these estimates compare to historical records can then be discussed here in the Discussion.

Answer: Tranferred to Results section

* Lines 173-181 somewhat explain the connection between the papers and the skeleton, but because this is not clear in the results section it is confusing to the reader early on.    This connection needs to be made a lot clearer.

Answer: The stamp image is a pivotal evidence in linking Estonian region for searches in archival information about the putative issuer of the document and linking the document with putative individuals as mentioned in the text. Text modified.

* Make a more succinct link between your discussion of surnames and non-paternity, and a clear line as to why it is important in this particular case.

Answer: The surnames in the Fig 5 are originating from genealogic information in the particular case.

* Lines 229-232. Make a clearer statement here about why Y-STR mutations rates do not explain the non-paternity result for Testee 2.

Answer: The statement of Y-STR mutations is provided.

 Conclusion

 * Although the authors suggest that these remains have been positively identified – did they also explore the possibility of these bones belonging to any other missing soldiers from the same paternal lineage (with a concordant Y-haplotype)??

Answer: Authors are confident based on integrated information that we were able to identify the particular soldier eliminating other soldier(s) from the same paternal lineage.

* Line 249.   What does “Another pedigree was questionable” mean?   Are the authors referring to the Testee 2 assumed pedigree?

Answer: The Y-STR results pointed to mismatches in haplotypes concerning Testee 2 pedigree.

* Rephrase the closing sentence – it does not make sense.

Answer: Agree, corrected.

Tables

 * Table 1 – what is the source of these metrics/criteria?  Cite them here.

Answer: Agree, corrected.

* Table 2A – make sure the reference formatting is consistent for then all in the first column in the table.

Answer: Checked and line 4 corrected.

Reviewer 3 Report

Dear authors, 

this is an interesting case study and I highly appreciate the interdisciplinarity of the research. Also, taking into consideration that many countries have similar issues I think that it would be interesting for the readers of Genes. 

Taking that into consideration, I think that the manuscript must be improved to answer some questions, that, in my opinion, were not answered in the text.

Introduction:

In lines 45 and 47 please cite the documents. 

In line 49, please explain and support by numbers the statement that the corresponding law did not improve the situation (How many were identified since the implementation of the law? How many graves were excavated?)

Materials and Methods

In the section Exhumation of remains, it is not clear if this research team did the excavations, too. Or the excavation was done by some other team? If it was done by the authors please explain how did you date the grave (How did you know that it is not from some other conflict?) If it was done by another excavation team, please cite them. Also, please describe all the grave findings, for example in the Discussion part are for the first time mentioned cartridges.

Section 2.2. should be renamed into  Anthropological analysis. For this part it is not methodologically correct to choose two formulas for stature estimation, thus the authors should choose only one. Also, I recommend using the DATA COLLECTION PROCEDURES FOR FORENSIC SKELETAL MATERIAL 2.0 for the measurements. 

Section 2.3. Did the authors work on the conservation? If so, please either cite the existing methodology you used, or if this is your own (unpublished methodology) please explain it in more detail (this listing in brackets is not understandable and confusing). If you did not do the analysis please cite the research. 

In section 2.4. please cite the church and statistics documents from national archives that you used. 

In section 2.5. - Did you use PowerPlex Y 23? If you did, please write so, instead of identifying all the loci themselves. For AB 3500 XL give the full name of the manufacturer. For YHRD give the full name of the abbreviation.

Results:

In section 3.1. What morphological features were examined for age estimation? At this part, I suggest that you either describe the remains (what bones were preserved?) or give a photo of the remains: Also, did you estimate age on clavicle union? Or some other features? I strongly recommend that you use Fordisc (if applicable) for both sex and age estimation. The Tables in the appendix are unnecessary and have no place in the manuscript. As the authors, you should decide if sex is male, female, or inconclusive, and not only list the features that you have examined. It is not clear if the rating was done by one or two researchers; if two, how did you get an agreement? If one, did you rate twice, or only once? I could not find the paper of Garmus and Jankauskas - but I presume that if they made the regression equations then they probably have the accuracy level provided. Methodologically speaking, the authors should choose only one (single) measurement or only one (regression equation for several measurements) which is the most accurate for the sex estimation. Giving the table of osteometric traits where several measurements were taken (and with different published work!) has no meaning and is additionally confusing. So, the authors can probably estimate sex using either regression equations or morphological traits and have the accuracy level. They can use also Fordisc for this. I think that they should have examined the other features of the biological profile (trauma analysis would be interesting). This is not even mentioned in the paper. 

Section 3.2. The documents in Figures 2 and 3 should be transcribed (the visible words) and written in the original language as well as the English translation. Also, it is not well explained how did you connect the documents from the grave to the exact document from the archive. For Figure 4, please provide the original language and English transcript. This is essential as this is a significant part of the quest for the person identified in this study. 

Before section 3.3. a section with genealogical data should be written, which is now in the Discussion part, and it does not belong to it. 

Section 3.3. Whose sample is a female reference sample?  If the sex of the skeleton could not be estimated anthropologically why did not you do amelogenin analysis? 

Discussion: 

Sections 155 to 172 belong to the results. The methodology here should be changed. Why calculating stature and body weight using two equations?  You should choose the one that is most appropriate for their sample (time frame and population)? And you can not confirm that the remains are "probably male" because the "probable female" is higher than the average female from that time and place. You have to decide if this is male, female or sex could not be estimated. Although I do not know what is preserved, from the tables in the appendix I can conclude that you probably have enough for sex estimation. 

Lines 179-181. It is still not clear to me how did you decide that you are searching for Jaan Lotus and not for the other two people (one of them, female, could be excluded based on the biological profile)? And, this part also belongs to the results. 

Lines 187-188 need citation of the archives. 

Author Response

Introduction:

In lines 45 and 47 please cite the documents. 

Answer: References added.

In line 49, please explain and support by numbers the statement that the corresponding law did not improve the situation (How many were identified since the implementation of the law? How many graves were excavated?)

Answer: Since 1925 identifications of exhumed remains were not performed.  

Materials and Methods

In the section Exhumation of remains, it is not clear if this research team did the excavations, too. Or the excavation was done by some other team? If it was done by the authors please explain how did you date the grave (How did you know that it is not from some other conflict?) If it was done by another excavation team, please cite them. Also, please describe all the grave findings, for example in the Discussion part are for the first time mentioned cartridges.

Answer: Exhumation was performed and leaded by our team member Arnold Unt.  

Section 2.2. should be renamed into  Anthropological analysis. For this part it is not methodologically correct to choose two formulas for stature estimation, thus the authors should choose only one. Also, I recommend using the DATA COLLECTION PROCEDURES FOR FORENSIC SKELETAL MATERIAL 2.0 for the measurements. 

Answer: Agreed and subtitle renamed.

Section 2.3. Did the authors work on the conservation? If so, please either cite the existing methodology you used, or if this is your own (unpublished methodology) please explain it in more detail (this listing in brackets is not understandable and confusing). If you did not do the analysis please cite the research. 

Answer: Corrected, as existing methodology (with references) was used.

In section 2.4. please cite the church and statistics documents from national archives that you used. 

Answer: Citations added.

In section 2.5. - Did you use PowerPlex Y 23? If you did, please write so, instead of identifying all the loci themselves. For AB 3500 XL give the full name of the manufacturer. For YHRD give the full name of the abbreviation.

Results:

In section 3.1. What morphological features were examined for age estimation? At this part, I suggest that you either describe the remains (what bones were preserved?) or give a photo of the remains: Also, did you estimate age on clavicle union? Or some other features? I strongly recommend that you use Fordisc (if applicable) for both sex and age estimation. The Tables in the appendix are unnecessary and have no place in the manuscript. As the authors, you should decide if sex is male, female, or inconclusive, and not only list the features that you have examined. It is not clear if the rating was done by one or two researchers; if two, how did you get an agreement? If one, did you rate twice, or only once? I could not find the paper of Garmus and Jankauskas - but I presume that if they made the regression equations then they probably have the accuracy level provided. Methodologically speaking, the authors should choose only one (single) measurement or only one (regression equation for several measurements) which is the most accurate for the sex estimation. Giving the table of osteometric traits where several measurements were taken (and with different published work!) has no meaning and is additionally confusing. So, the authors can probably estimate sex using either regression equations or morphological traits and have the accuracy level. They can use also Fordisc for this. I think that they should have examined the other features of the biological profile (trauma analysis would be interesting). This is not even mentioned in the paper. 

Answer: Comments revisited.

Section 3.2. The documents in Figures 2 and 3 should be transcribed (the visible words) and written in the original language as well as the English translation. Also, it is not well explained how did you connect the documents from the grave to the exact document from the archive. For Figure 4, please provide the original language and English transcript. This is essential as this is a significant part of the quest for the person identified in this study.

Answer: Not agreed, as the visible documents do not add additional information in translation.  

Before section 3.3. a section with genealogical data should be written, which is now in the Discussion part, and it does not belong to it. 

Answer: We would like to leave the genealogical data in the Discussion part for the better understanding.

Section 3.3. Whose sample is a female reference sample?  If the sex of the skeleton could not be estimated anthropologically why did not you do amelogenin analysis? 

Answer: Female sample was collected based on genealogical data for sake of analysis of autosomal markers, however in the current context it is not relevant, so the text is corrected.

Answer:

Discussion: 

Sections 155 to 172 belong to the results. The methodology here should be changed. Why calculating stature and body weight using two equations?  You should choose the one that is most appropriate for their sample (time frame and population)? And you can not confirm that the remains are "probably male" because the "probable female" is higher than the average female from that time and place. You have to decide if this is male, female or sex could not be estimated. Although I do not know what is preserved, from the tables in the appendix I can conclude that you probably have enough for sex estimation. 

Answer: The text and appendixes has been clarified.

Lines 179-181. It is still not clear to me how did you decide that you are searching for Jaan Lotus and not for the other two people (one of them, female, could be excluded based on the biological profile)? And, this part also belongs to the results. 

Answer: We involved individuals who provided reference samples for identification proposes based on integrated information.

Round 2

Reviewer 1 Report

The authors corrected the manuscript appropriately according to my suggestion.

The manuscript was improved.

Author Response

Introduction:

  1. Comment first review: In lines 45 and 47 please cite the documents. 

Answer: References added.

Comment second review: Thank you for resolving the comment, and please renumber the references in this section – references 24-26 should be references 3-5.

Answer: References added and renumbered.

Materials and Methods

  1. Comment first review: In the section Exhumation of remains, it is not clear if this research team did the excavations, too. Or the excavation was done by some other team? If it was done by the authors please explain how did you date the grave (How did you know that it is not from some other conflict?) If it was done by another excavation team, please cite them. Also, please describe all the grave findings, for example in the Discussion part are for the first time mentioned cartridges.

Answer: Exhumation was performed and leaded by our team member Arnold Unt.  

Comment second review: please describe the grave findings at this section. It was also a question in my previous review.

Answer: Agreed and description added. 

  1. Comment first review: Section 2.2. should be renamed into  Anthropological analysis. For this part it is not methodologically correct to choose two formulas for stature estimation, thus the authors should choose only one. Also, I recommend using the DATA COLLECTION PROCEDURES FOR FORENSIC SKELETAL MATERIAL 2.0 for the measurements. 

Answer: Agreed and subtitle renamed.

Comment second review: Still not clear what methodology for bone measurements was chosen. The paper you are citing is not available to me. Also, the authors of that paper probably also used some of the standard skeletal measurements. Please, amend.

Answer: Agreed and additional references used in bone analysis included.

  1. Comment first review: Section 2.3. Did the authors work on the conservation? If so, please either cite the existing methodology you used, or if this is your own (unpublished methodology) please explain it in more detail (this listing in brackets is not understandable and confusing). If you did not do the analysis please cite the research. 

Answer: Corrected, as existing methodology (with references) was used.

Comment second review: still not explained the conservation process, one of the documents cited is the ethics document, and although it is useful and necessary it does not offer the methodology of the conservation. This section should be written as the DNA analysis section was written, so that the methodology in similar cases can be repeated.

Answer: Agreed and text amended.  

  1. Comment first review: In section 2.5. - Did you use PowerPlex Y 23? If you did, please write so, instead of identifying all the loci themselves. For AB 3500 XL give the full name of the manufacturer. For YHRD give the full name of the abbreviation.

Comment second review: This question remained unanswered, please amend.

Answer: Agreed and text amended and clarified.

Results:

  1. Comment first review: In section 3.1. What morphological features were examined for age estimation? At this part, I suggest that you either describe the remains (what bones were preserved?) or give a photo of the remains: Also, did you estimate age on clavicle union? Or some other features? I strongly recommend that you use Fordisc (if applicable) for both sex and age estimation. The Tables in the appendix are unnecessary and have no place in the manuscript. As the authors, you should decide if sex is male, female, or inconclusive, and not only list the features that you have examined. It is not clear if the rating was done by one or two researchers; if two, how did you get an agreement? If one, did you rate twice, or only once? I could not find the paper of Garmus and Jankauskas - but I presume that if they made the regression equations then they probably have the accuracy level provided. Methodologically speaking, the authors should choose only one (single) measurement or only one (regression equation for several measurements) which is the most accurate for the sex estimation. Giving the table of osteometric traits where several measurements were taken (and with different published work!) has no meaning and is additionally confusing. So, the authors can probably estimate sex using either regression equations or morphological traits and have the accuracy level. They can use also Fordisc for this. I think that they should have examined the other features of the biological profile (trauma analysis would be interesting). This is not even mentioned in the paper. 

Comment second review: The comments are partly resolved. The issues that need resolving re as follows: The Tables in the appendix are unnecessary and have no place in the manuscript. As the authors, you should decide if sex is male, female, or inconclusive, and not only list the features that you have examined. It is not clear if the rating was done by one or two researchers; if two, how did you get an agreement? If one, did you rate twice, or only once? I could not find the paper of Garmus and Jankauskas - but I presume that if they made the regression equations then they probably have the accuracy level provided. Methodologically speaking, the authors should choose only one (single) measurement or only one (regression equation for several measurements) which is the most accurate for the sex estimation. Giving the table of osteometric traits where several measurements were taken (and with different published work!) has no meaning and is additionally confusing. So, the authors can probably estimate sex using either regression equations or morphological traits and have the accuracy level. They can use also Fordisc for this. I think that they should have examined the other features of the biological profile (trauma analysis would be interesting). This is not even mentioned in the paper.

Answer: Agreed and text amended through additional references used in bone measurements. The individual variability of physical traits is the main concern in the context of particular bone measurements.    

  1. Comment first review: Section 3.2. The documents in Figures 2 and 3 should be transcribed (the visible words) and written in the original language as well as the English translation. Also, it is not well explained how did you connect the documents from the grave to the exact document from the archive. For Figure 4, please provide the original language and English transcript. This is essential as this is a significant part of the quest for the person identified in this study.

Answer: Not agreed, as the visible documents do not add additional information in translation.  

Comment second review: If the authors believe that “the visible documents do not add additional information in translation” than they should remove them from the manuscript.

Answer: Seems that lost in translation. The translation from Estonian to English on the visible documents in Figure 2 and 3 is provided in the description. The main aim of providing these figures is related to Document conservation capabilities from aged materials to aid identification of remains.     

Discussion: 

  1. Comment first review: Sections 155 to 172 belong to the results. The methodology here should be changed. Why calculating stature and body weight using two equations?  You should choose the one that is most appropriate for their sample (time frame and population)? And you can not confirm that the remains are "probably male" because the "probable female" is higher than the average female from that time and place. You have to decide if this is male, female or sex could not be estimated. Although I do not know what is preserved, from the tables in the appendix I can conclude that you probably have enough for sex estimation. 

Answer: The text and appendixes has been clarified.

Comment second review: Still not clear did you use reference 7 or 8 for the calculation? And sill the calculation makes no sense – what are you proving with this – that it is either male either female? This should be left out from the paper. As I have already stated _ And you can not confirm that the remains are "probably male" because the "probable female" is higher than the average female from that time and place. Also, the text (same) about the calculations is repeated in results and discussion.

Answer: We would like to point that in identification (old) remains the integrated information from different sources using different scientific fields are at utmost importance. However, we left out the second calculation method. The repeating text part removed from Discussion.

Reviewer 3 Report

Dear Authors,

Although some of the corrections were made, the manuscript still needs improvements as some of the changes are not implemented into the manuscript. The comments are below. 

Introduction:

  1. Comment first review: In lines 45 and 47 please cite the documents. 

Answer: References added.

Comment second review: Thank you for resolving the comment, and please renumber the references in this section – references 24-26 should be references 3-5.

  1. Comment first review: In line 49, please explain and support by numbers the statement that the corresponding law did not improve the situation (How many were identified since the implementation of the law? How many graves were excavated?)

Answer: Since 1925 identifications of exhumed remains were not performed.  

Thank you for resolving the comment.

Materials and Methods

  1. Comment first review: In the section Exhumation of remains, it is not clear if this research team did the excavations, too. Or the excavation was done by some other team? If it was done by the authors please explain how did you date the grave (How did you know that it is not from some other conflict?) If it was done by another excavation team, please cite them. Also, please describe all the grave findings, for example in the Discussion part are for the first time mentioned cartridges.

Answer: Exhumation was performed and leaded by our team member Arnold Unt.  

Comment second review: please describe the grave findings at this section. It was also a question in my previous review.

  1. Comment first review: Section 2.2. should be renamed into  Anthropological analysis. For this part it is not methodologically correct to choose two formulas for stature estimation, thus the authors should choose only one. Also, I recommend using the DATA COLLECTION PROCEDURES FOR FORENSIC SKELETAL MATERIAL 2.0 for the measurements. 

Answer: Agreed and subtitle renamed.

Comment second review: Still not clear what methodology for bone measurements was chosen. The paper you are citing is not available to me. Also, the authors of that paper probably also used some of the standard skeletal measurements. Please, amend.

  1. Comment first review: Section 2.3. Did the authors work on the conservation? If so, please either cite the existing methodology you used, or if this is your own (unpublished methodology) please explain it in more detail (this listing in brackets is not understandable and confusing). If you did not do the analysis please cite the research. 

Answer: Corrected, as existing methodology (with references) was used.

Comment second review: still not explained the conservation process, one of the documents cited is the ethics document, and although it is useful and necessary it does not offer the methodology of the conservation. This section should be written as the DNA analysis section was written, so that the methodology in similar cases can be repeated.

  1. Comment first review: In section 2.4. please cite the church and statistics documents from national archives that you used. 

Answer: Citations added.

Comment second review: Thank you for resolving the comment.

  1. Comment first review: In section 2.5. - Did you use PowerPlex Y 23? If you did, please write so, instead of identifying all the loci themselves. For AB 3500 XL give the full name of the manufacturer. For YHRD give the full name of the abbreviation.

Comment second review: This question remained unanswered, please amend.

Results:

  1. Comment first review: In section 3.1. What morphological features were examined for age estimation? At this part, I suggest that you either describe the remains (what bones were preserved?) or give a photo of the remains: Also, did you estimate age on clavicle union? Or some other features? I strongly recommend that you use Fordisc (if applicable) for both sex and age estimation. The Tables in the appendix are unnecessary and have no place in the manuscript. As the authors, you should decide if sex is male, female, or inconclusive, and not only list the features that you have examined. It is not clear if the rating was done by one or two researchers; if two, how did you get an agreement? If one, did you rate twice, or only once? I could not find the paper of Garmus and Jankauskas - but I presume that if they made the regression equations then they probably have the accuracy level provided. Methodologically speaking, the authors should choose only one (single) measurement or only one (regression equation for several measurements) which is the most accurate for the sex estimation. Giving the table of osteometric traits where several measurements were taken (and with different published work!) has no meaning and is additionally confusing. So, the authors can probably estimate sex using either regression equations or morphological traits and have the accuracy level. They can use also Fordisc for this. I think that they should have examined the other features of the biological profile (trauma analysis would be interesting). This is not even mentioned in the paper. 

Answer: Comments revisited.

Comment second review: The comments are partly resolved. The issues that need resolving re as follows: The Tables in the appendix are unnecessary and have no place in the manuscript. As the authors, you should decide if sex is male, female, or inconclusive, and not only list the features that you have examined. It is not clear if the rating was done by one or two researchers; if two, how did you get an agreement? If one, did you rate twice, or only once? I could not find the paper of Garmus and Jankauskas - but I presume that if they made the regression equations then they probably have the accuracy level provided. Methodologically speaking, the authors should choose only one (single) measurement or only one (regression equation for several measurements) which is the most accurate for the sex estimation. Giving the table of osteometric traits where several measurements were taken (and with different published work!) has no meaning and is additionally confusing. So, the authors can probably estimate sex using either regression equations or morphological traits and have the accuracy level. They can use also Fordisc for this. I think that they should have examined the other features of the biological profile (trauma analysis would be interesting). This is not even mentioned in the paper.

  1. Comment first review: Section 3.2. The documents in Figures 2 and 3 should be transcribed (the visible words) and written in the original language as well as the English translation. Also, it is not well explained how did you connect the documents from the grave to the exact document from the archive. For Figure 4, please provide the original language and English transcript. This is essential as this is a significant part of the quest for the person identified in this study.

Answer: Not agreed, as the visible documents do not add additional information in translation.  

Comment second review: If the authors believe that “the visible documents do not add additional information in translation” than they should remove them from the manuscript.

  1. Comment first review: Before section 3.3. a section with genealogical data should be written, which is now in the Discussion part, and it does not belong to it. 

Answer: We would like to leave the genealogical data in the Discussion part for the better understanding.

Comment second review: If the authors consider this better, I agree.

  1. Comment first review: Section 3.3. Whose sample is a female reference sample?  If the sex of the skeleton could not be estimated anthropologically why did not you do amelogenin analysis? 

Answer: Female sample was collected based on genealogical data for sake of analysis of autosomal markers, however in the current context it is not relevant, so the text is corrected.

Comment second review: Thank you for resolving this.

Discussion: 

  1. Comment first review: Sections 155 to 172 belong to the results. The methodology here should be changed. Why calculating stature and body weight using two equations?  You should choose the one that is most appropriate for their sample (time frame and population)? And you can not confirm that the remains are "probably male" because the "probable female" is higher than the average female from that time and place. You have to decide if this is male, female or sex could not be estimated. Although I do not know what is preserved, from the tables in the appendix I can conclude that you probably have enough for sex estimation. 

Answer: The text and appendixes has been clarified.

Comment second review: Still not clear did you use reference 7 or 8 for the calculation? And sill the calculation makes no sense – what are you proving with this – that it is either male either female? This should be left out from the paper. As I have already stated _ And you can not confirm that the remains are "probably male" because the "probable female" is higher than the average female from that time and place. Also, the text (same) about the calculations is repeated in results and discussion.

  1. Comment first review: Lines 179-181. It is still not clear to me how did you decide that you are searching for Jaan Lotus and not for the other two people (one of them, female, could be excluded based on the biological profile)? And, this part also belongs to the results. 

Answer: We involved individuals who provided reference samples for identification proposes based on integrated information.

Comment second review: Thank you for resolving this.

Author Response

(The authors gave the same response as above.)
